# Patterns and trends of the dominant environmental controls of net biome productivity

Barbara Marcolla[1], Mirco Migliavacca[2] , Christian Rödenbeck[2], Alessandro Cescatti[3]

*Correspondence to*: Alessandro Cescatti (Alessandro.CESCATTI@ec.europa.eu)

1 Sustainable Agro-ecosystems and Bioresources Department, IASMA Research and Innovation Centre, Fondazione Edmund Mach Via E. Mach 1, 38010 San Michele all'Adige, (TN), Italy
2 Max Planck Institute for Biogeochemistry, Jena, 07745, Germany
3 European Commission, Joint Research Centre (JRC), Ispra, Italy

**Abstract.** In the last decades terrestrial ecosystems have reabsorbed on average more than one quarter of anthropogenic emissions (Le Quéré et al., 2018). However, this large carbon sink is modulated by climate and is therefore highly variable in time and space. The magnitude and temporal changes in the sensitivity of terrestrial $CO_2$ fluxes to climate drivers are key factors to determine future atmospheric $CO_2$ concentration and climate trajectories. In the literature, there is so far a strong focus on the climatic controls of daily and long-term variability, while less is known about the key drivers at seasonal time scale and about their variation over time (Wohlfahrt et al., 2008). This latter temporal scale is relevant to assess which climatic drivers dominate the seasonality of the fluxes and to understand which factors limit the $CO_2$ exchange during the course of the year. Here, we investigate the global sensitivity of net terrestrial $CO_2$ fluxes, derived from atmospheric inversion, to three key climate drivers (i.e. global radiation and temperature from WFDEI, soil water content from ERA-Interim) from weekly to seasonal temporal scales, in order to explore the short-term interdependence between climate and the terrestrial carbon budget. We observed that the $CO_2$ exchange is controlled by temperature during the carbon uptake period over most of the land surface (from 55 to 52% of the total surface), while radiation is the most widespread dominant climate driver during the carbon release period (from 64 to 70% of the total surface). As expected, soil water content plays a key role in arid regions of the Southern Hemisphere both during the carbon uptake and the carbon release period. Looking at the decadal trend of these sensitivities (1985-2016) we observed that the importance of radiation as a driver is increasing over time, while we observed a decrease in sensitivity to temperature in Eurasia. Overall, we show that flux temporal variation due to a specific driver has been dominated by the temporal changes in ecosystem sensitivity (i.e. the response of ecosystem to climate) rather than to the temporal variability of the climate driver itself over the last decades. Ultimately, this analysis shows that the ecosystem response to climate is significantly changing both in space and in time, with potential repercussion on the future terrestrial $CO_2$ sink and therefore on the role that land may play in climate trajectories.

## 1 Introduction

Just over one quarter of the anthropogenic emissions of carbon dioxide ($CO_2$) on average are reabsorbed by terrestrial ecosystems (Le Quéré et al., 2018). This large sink is influenced by climate and therefore by its short- and long-term variability (Beer et al., 2010; Ciais et al., 2005; Rödenbeck et al., 2018; Sitch et al., 2015). In fact, key climate drivers, like radiation, temperature, precipitation regime and soil moisture, control the fundamental processes of photosynthesis and respiration that are modulating the net ecosystem $CO_2$ exchange (Reich et al., 2018). Moreover, climate change is affecting the phenological cycle of plants and, therefore, the functioning of ecosystems which in turn affect climate (Richardson et al., 2013). Due to this interrelation, model studies show that the response of land $CO_2$ fluxes to climate drivers may heavily determine the future climate trajectories (Friedlingstein et al., 2001). Ultimately, the large uncertainty of climate projections could be significantly improved with a better understanding of vegetation response to the climate variability observed in the past (Papagiannopoulou et al., 2017).

In the last decades the climate sensitivity of terrestrial ecosystem $CO_2$ exchange has been investigated at different temporal and spatial scales and with a variety of measurement techniques ranging from eddy covariance, which continuously monitor fluxes at local scale (Baldocchi, 2003; Baldocchi et al., 2001), to global scale retrievals based on satellite remote sensing.

An increasing range of sensors on different satellite platforms are continuously monitoring the structural and functional properties of global vegetation with different techniques and wavebands (i.e. optical, thermal, microwave, etc.). The combination of multiple sources of Earth observations have proved to be a valuable method to assess land-climate interactions at large scale and to constrain model representation (Alkama and Cescatti, 2016; Duveiller et al., 2018; Jung et al., 2017; Ryu et al., 2019; Tramontana et al., 2016).

Evidence-driven model products based on data assimilation are another important tool to analyse the vegetation-climate interplay and can be used to assess the generalizability of ground-based observations (Fernández-Martínez et al., 2019). Among these, atmospheric inversions (such as the Jena CarboScope Inversion used here) combine modelled atmospheric transport with high precision measurements of atmospheric $CO_2$ concentrations to derive surface fluxes (Rödenbeck et al., 2003). Atmospheric inversions are particularly suitable for the assessment of vegetation-climate interactions because these data-products are not assuming a-priori any trend in the inter-play between climate and fluxes. Besides, inversions provide global data over several decades and are therefore useful to assess temporal changes at large spatial scale.

The sensitivity of ecosystem net biome productivity (NBP) to climate variability has been so far mostly investigated at annual scale, while it is still poorly investigated across multiple sub-annual temporal scales. However, it is at the diurnal to seasonal scales where climate variability is directly acting on ecosystems (e.g., though heat waves, droughts, or cold spells) (Katul et al., 2001), while annual anomalies are just the sum over such sub-annual responses. Besides, climate variability can have different impacts on the $CO_2$ flux (enhancing or dampening its variability) depending on the time period of the year when it occurs (Marcolla et al., 2011; Sippel et al., 2016). Thus, it is important to assess the limiting climate factors that control weekly or monthly evolution of ecosystem carbon fluxes in order to assess the vulnerability and forecast the future evolution of the

ecosystem carbon budgets (De Keersmaecker et al., 2015; le Maire et al., 2010). To this scope, in our work we explore the recent patterns and temporal trends of the environmental drivers of NBP. In particular, we assess the relative importance of

key drivers like global radiation, temperature and soil water content (Jung et al., 2017) at the sub-annual time scales (weekly to seasonal). The analysis was framed to i) identify the limiting factors of global net biome productivity (NBP) from weekly to seasonal time scales ii) assess how the NBP sensitivity to the main climate drivers has been changing in the recent decades and iii) quantify the contribution of the variations in the climate drivers and in the response of ecosystems to climate in determining the total temporal variability of $CO_2$ fluxes.

**2 Materials and Methods**

**2.1 Datasets**

Gridded global flux estimates were obtained from the top-down product Jena CarboScope $CO_2$ Inversion (version s85_v4.1, 21 atmospheric sites) (Rödenbeck et al., 2003). Atmospheric inversions yield surface flux fields that achieve the best match to high-precision measurements of atmospheric $CO_2$ concentrations, where the fluxes are linked to atmospheric mole fractions

by modelled atmospheric transport. For this specific inversion, atmospheric transport is simulated by the global three-dimensional transport model TM3 (Heimann and Körner, 2003) driven by meteorological data from the NCEP reanalysis (Kalnay et al., 1996). The product version used in this analysis covers the period 1985-2016 at daily time scale; however, since the inversion uses temporal a-priori correlations that smooth away any flux variations faster than about a week, the minimum time resolution we analysed is 7 days. The Jena Inversion is particularly suited for the analysis of temporal trends and

variability since it is based on a temporally constant observation network for the entire simulation period, in order to minimize spurious influences from the beginning or ending of data records on the spatio-temporal variation of the fluxes. Among the versions of the Jena CarboScope $CO_2$ Inversion we selected s85_v4.1 since it represent a good compromise between the length of the time series (needed to assess temporal trends) and the density of the observation network (required to have a good spatial representativeness of the dataset). In order to prove the robustness of the results we performed part of the analysis also with

other versions of the s85 Jena Carboscope product that were produced to explore the uncertainty of the inversion driven by the priors and by the spatio-temporal correlation of the error (s85oc_tight_v4.3 with halved prior uncertainty, s85oc_loose_v4.3 with double prior uncertainty, s85oc_short_v4.3 with shorter spatial correlation and s85oc_fast_v4.3 with shorter temporal correlation). In addition, we explored also a different version of the product (s81oc_v4.3). We limited the analysis of the uncertainty to different versions of the Jena CarboScope $CO_2$ Inversion since, to our knowledge, all others long-term inversions

are produced with a varying observation network (the number of atmospheric stations used in the inversion is changing during the time series) and are therefore not adequate for the scope of our study.

Concerning climate variables, global radiation (RG), air temperature (TA) and soil water content (SWC) were used as  key drivers for NBP (Jung et al., 2017; Ma et al., 2007; Papagiannopoulou et al., 2017). These environmental variables are generally recognized as the major factors driving the variation of $CO_2$ fluxes from hourly to multi-day time scale (Chu et al., 2016;

Richardson et al., 2007), while the response at longer time scales becomes more complex and often involves indirect effects through functional changes (Teklemariam et al., 2010). VPD has been evaluated as an alternative to SWC, but ultimately it was not included in the analysis since it controls only one of the two processes (gross primary productivity, GPP) determining the ecosystem NBP, while SWC has an impact on both GPP and total ecosystem respiration (TER).

Global radiation and air temperature data were retrieved from the WFDEI database (Weedon et al., 2014). The dataset covers
the period 1985-2016 with a spatial resolution of 0.5°x0.5° and a temporal resolution of 1 day. The WFDEI meteorological forcing data set has been generated using the same methodology as the WATCH Forcing Data (WFD) by making use of the ERA-Interim reanalysis data. ERA-Interim is a global atmospheric reanalysis from 1979, continuously updated in real time by the European Centre for Medium-Range Weather Forecasts (ECMWF, (Berrisford et al., 2011). The ERA-Interim dataset was also used to retrieve the soil water content (level 2, from 0.07 to 0.28 m depth).

**2.2 Statistical data analysis**

All datasets were aggregated at the spatial resolution of the inversion product (5°x3.75°) with the R package "raster" using the mean of the variables as aggregation function (Hijmans, 2017). A moving window of 7, 30, 90 days was then applied to the data to have data at weekly, monthly and seasonal temporal resolution, respectively. We chose this statistical method among those available (e.g. Fourier transformation, wavelet analysis, SSA, etc.) because of a combination of simplicity, robustness
and clarity.

Multi-linear regression models have been extensively used to assess the inter-linkages between global vegetation and climate (Barichivich et al., 2014; Nemani et al., 2003). In this study regressions between Jena Carboscope NBP and global radiation (RG), air temperature (TA) and soil water content (SWC) were estimated at pixel level using the R package "glmnet" (Friedman et al., 2010), which is suitable to calculate linear regression coefficients in case of collinearity, as it is often the case
with multiple climate drivers. The presence of collinearity was assessed computing the variance inflation factor (Figure S1), which measures how much the variance of a regression coefficient is inflated due to multi-collinearity in the model (Gareth et al., 2014). When multi-collinearity occurs, least squares estimates are unbiased, but their variances are so large that they may be completely inaccurate. Hence, to account for collinearity the loss function is modified in a way that not only the sum of squared residuals is minimized, but also the size of parameter estimates is penalized, in order to shrink them towards zero. The
ideal penalty is somewhere in between 0 (ordinary least square) and $\infty$ (all coefficients shrunk to 0) and gives the minimum mean cross-validated error.

Regression coefficients for each pixel were estimated first using the entire time series, and then separately for the Carbon Uptake Period (CUP, defined as the period when the land acts as a carbon sink since gross primary productivity dominates over respiratory terms) and the Carbon Release Period (CRP, when respiration is larger than gross primary productivity and
the land is a carbon source). Since GPP and TER cannot be derived from inversion products, we performed the regression

analysis using NBP of CUP and of CRP as proxies of GPP and TER, respectively (Migliavacca et al., 2011, 2015). Climatological CUP and CRP were identified using the seasonality of NBP (sign convention: NBP>0 corresponds to uptake) for each pixel, periods with NBP>0 were classified as CUP and periods with NBP<0 as CRP.

The absolute value of standardized coefficients was used as a measure of the relative importance of the drivers. Hence the dominant driver for each pixel was the one having the largest coefficient. In order to assess the temporal variation of the sensitivity to climate drivers, the observation period was split into 8 sub-periods of 4 years each. For each sub-period a multi-linear regression of NBP versus the selected climate drivers (RG, TA, SWC) was estimated at pixel level, obtaining 8 angular coefficients (i.e. sensitivities) for each driver ($m_{driver}$). Average values of the drivers were also calculated for each sub-period. The temporal trend of the sensitivities to climate drivers was investigated with linear regressions versus time at pixel level.

The contributions to NBP total temporal variability due to temporal variation in the climate drivers and in the ecosystem sensitivity-to-drivers were separately estimated according to the following equation:

$$\frac{dNBP}{dt}\bigg|_{driver} = \frac{dm_{driver}}{dt}driver(t) + m_{driver}(t)\frac{d(driver)}{dt}$$

where $m_{driver}$ is the coefficient of a driver in the multi-linear regression.

The contribution of the temporal change in the ecosystem sensitivity-to-driver ($\frac{dm_{driver}}{dt}driver(t)$) was obtained estimating a linear regression against time for the 8 angular coefficients ($m_{driver}$) previously calculated for the 8 sub-periods of 4 years, and the temporal sensitivity obtained from the regression ($\frac{dm_{driver}}{dt}$) was multiplied by the average value of the driver in the sub-periods. The contribution of the temporal changes in the drivers ($m_{driver}(t)\frac{d(driver)}{dt}$) was obtained estimating a linear regression of the sub-periods' average driver values against time, and the temporal sensitivity of the driver was multiplied by the sensitivity-to-driver coefficients.

## 3 Results

### 3.1 Dominant drivers across regions and climates

The analysis of the drivers of sub-annual NBP fluctuations shows clear spatial patterns, where single climate variable dominates specific geographic regions in the different climate zones (Figure 1). In particular, the climate driver that controls the fluctuation of NBP in most of the Northern Hemisphere is radiation, with an increasing dominance from the weekly to seasonal temporal scale, while in the Southern Hemisphere soil water content controls NBP in the driest regions of Africa and South America, and radiation and temperature dominate elsewhere.

Looking at the sign of the relationships between NBP and drivers, it is interesting to notice that the global maps are dominated by positive correlations between drivers and NBP (regions with + sign in Figure 1), meaning that the terrestrial land sink is larger during periods with higher temperature, radiation and soil water content. As expected, negative correlation with radiation occurs in tropical regions where high radiation loads are related to stressful conditions (e.g. heat stress, water limitation, or a combination of the two).

Looking at the differences between temporal scales, we observe that the area with positive correlation is rather stable at the various time resolution (11% increase from 7 to 90 days), whereas the areas with negative correlations show a much stronger increase (50% increase from 7 to 90 days), suggesting that the negative interplay between radiation and NBP typically occurs at longer timesteps than the positive one. This should be interpreted by considering that positive correlations are likely due to the direct effect of the rapid response of photosynthesis to light, whereas negative correlations are due to indirect effect on the overall growing conditions, typically leading to stomatal limitation (e.g. dry season in the tropical regions with high VPD and low soil water content).

Temperature is the second most frequent dominant variable and controls the tropics in the Northern Hemisphere, the southernmost latitudes and East Asia. Similarly to radiation, the effect of temperature on the weekly to seasonal variation in NBP is mostly positive (Wu et al., 2015) except in arid regions of the Middle East. Soil water content controls the boreal latitudes and has a negative effect on carbon fluxes (drier periods show higher uptake); while in arid regions of the Southern Hemisphere it has a positive effect (humid periods show higher uptake).

In order to assess the consistency of the results the analysis of the dominant climate controls was repeated with other 5 versions of the Carboscope inversion. Results confirm the robustness of the finding, with an agreement on the dominant driver in 5 out of 6 products over about 90% of the land surface (Figure S2, supplementary material).

## 3.2 Dominant drivers across temporal phases

Since the processes that dominate the $CO_2$ exchange are different between the period of carbon uptake when the land is a sink (CUP) and the period of carbon release (CRP), the regression analysis was repeated separately for these two phases of the ecosystem carbon budget (Figure 2, 3).

Results show that the dominant drivers of the high frequency fluctuation in NBP are different between the two periods. In the continental regions of the Boreal Hemisphere, the variability in the period dominated by photosynthesis (CUP) is mostly driven by positive relationships with temperature on NBP, while the temperate zone shows a mixed pattern of temperature and radiation limitation (Figure 2). During CUP in the Southern Hemisphere a key role is played by soil water availability which is positively correlated with NBP fluxes across the Tropical region.

Interesting results emerge from the analysis of the key drivers during the carbon release period (Figure 2, 3). Globally the most common limiting factor is radiation, with a strong positive and negative control in the Northern and Southern Hemisphere, respectively. This distinct pattern is related to the different processes limiting the carbon uptake in the two Hemispheres: the

low radiation load occurring off-season in the Boreal Hemisphere and the condition of high radiation and aridity in the Southern Hemisphere (confirmed by the positive effect of soil water content).

Altogether we observed an important asymmetry in the sign of the controlling drivers between the CUP and CRP. While the former is stimulated by the increase in the drivers on a large fraction of the Earth surface (more than 80% on average for all drivers), the latter shows a mixed pattern where the $CO_2$ sink is stimulated in about half of the planet and depressed in the

other half by radiation (Figure 3). Concerning temperature, the key parameter in a global change perspective, the asymmetry is even stronger, with a control overwhelmingly positive in the CUP (in ~85% of the temperature-dominated surface at all the investigated temporal resolutions) and mostly negative (in 76%, 68% and 45% of the temperature-dominated surface at 7, 30 and 90 days respectively) in the CRP. This pattern is likely due to the temperature stimulation of the two opposite processes GPP and TER that controls NBP during CUP and CRP, respectively.


**3.3 Temporal trends of environmental controls of NBP**

In a scenario of rapidly changing climate it is particularly important to assess how the sensitivity of NBP to the different drivers has been changing over time and in which geographic regions. To this end, Figure 4 summarizes global maps of the average values (left panels) and temporal trends (right panels) of the regression coefficients that can be ultimately interpreted as

sensitivities to climate drivers. Regressions have been computed at 7 days temporal resolution and for the all-year period. Concerning radiation, the positive sensitivity shown in Fig 1 and in Fig. 4a is increasing in time (Fig 4b) in most of the Northern Hemisphere. This positive trend observed in the last three decades is likely due to the increasing leaf area index (LAI) and primary productivity of the northern regions, leading to increased light use efficiency and therefore to a stronger control of NBP by light. On the contrary, in the Southern Hemisphere the average sensitivity to radiation is mostly negative and the

trends are heterogeneous, since light may exerts a negative indirect effect on the carbon budget in warm/arid climates. A mostly positive sensitivity of NBP to soil water content occurs in arid regions, where evapotranspiration is supply limited and water stress may limit productivity. On the contrary, in northern regions, where evaporation is limited by atmospheric demand, the sensitivity is negative (Fig 4e). The trend in the sensitivity to water availability does not show a clear spatial pattern, likely due to the complex interplay between changes in precipitation and evapotranspiration in the different regions.

Ultimately the sensitivity is likely to decrease where water availability is increasing, and vice versa it may increase in areas that are experiencing increasing water stress (Fig 4f).

This analysis was repeated with other five inversion products to check for result consistency. We observed a low standard deviation of NBP sensitivity to climate drivers among products (Figure S3, second column) and an overall agreement in terms of temporal trends of these sensitivities over most of the land surface (Figure S3, third column).

In order to explore the relationships between sensitivities, trends and background climate, results shown in the global maps of Figure 4 are summarized according to climate coordinates (i.e. annual cumulative rainfall and mean temperature, Figure 5). A

clear pattern is emerging for radiation, with negative sensitivities in regions with high and very low temperatures, independently from precipitation values, while at intermediate temperatures radiation has a consistent positive effect on carbon fluxes. Figure 5 shows that the climate dependence of the trend in sensitivity to radiation generally follows the pattern of the mean sensitivity, with positive trends in climate regions characterized by positive sensitivity and vice-versa. Ultimately this combination of mean effects and trends is increasing the spatial variance in the ecosystem response to light, amplifying the differences between regions with positive and negative controls.

Similarly to radiation, temperatures also show positive sensitivities at intermediate mean temperatures. However, the different patterns of temperature and radiation trends suggest that the underlying processes triggered by the two drivers are likely different. In fact, the climate dependence of the trend in sensitivity to temperature doesn't follow the pattern of the mean sensitivity, being opposite in sign at intermediate temperatures and leading in this way to a homogenization of its spatial variability. The sign and magnitude of the sensitivity of NBP to soil water content is clearly controlled by the background mean temperature, with a sharp threshold at about 7 °C between regions with positive and negative sensitivity. On average the sensitivity to soil water content is increasing in regions warmer than 0 °C, but with considerable local variation, suggesting in general an increasing impact of water limitations on the fluctuations of the terrestrial carbon cycle, as also reported by (Jung et al., 2017). It is interesting to highlight the positive trend of the soil water control in cold climates (temperature between -2 and 7 °C), where historically the mean signal has been negative. This finding is in agreement with the recent literature about the increasing control of soil water content on the NBP of boreal ecosystems (Buermann et al., 2018; Lian et al., 2020).

Analysing the trends of NBP sensitivity to climatic drivers separately for CUP and CRP (Figure 6) we noticed that the importance of radiation is increasing in most of the Northern Hemisphere in both periods, suggesting an overall increase in the occurrence of light-limited photosynthesis. This is likely due to a combination of warming, nitrogen deposition and $CO_2$ fertilization that have led to an extended growing season length and greening. In particular, the large increase in the sensitivity to radiation (likely related to the greening of the Planet, as suggested by the spatial patterns of LAI trends reported by Zhu et al., (2016)) dominates the radiation-related changes of NBP. The increase of light-limitation goes hand in hand with the decline of temperature limitation, in particular during the CRP in Eurasia. Opposite trends of sensitivity to radiation and temperature occur also in the Amazon, where during the CUP we observe an increasing control of radiation and decreasing control of temperature, and the opposite during CRP.

Finally, we factored out the observed total temporal variability in NBP in two components: the variability due to the temporal change in the drivers and that due to variations in the ecosystem response to drivers (i.e. ecosystem sensitivity to climate). Results show that the average contribution of the temporal change in sensitivity (Fig. 7 left column) is on average much larger than the contribution of the driver variability (Fig. 7 right column). This means that indirect climate effects, leading to a change of ecosystem sensitivity (e.g. aridity that increases the NBP sensitivity to water availability), are extremely relevant in determining the overall variability of the global NBC and may eventually amplify (when the two components have the same sign) or dampen (when opposite in sign) the effect of variation in climate drivers on terrestrial ecosystems.

## 4 Discussion

### 4.1 Potentials and limitations of the methodology

The analysis presented in this contribution largely builds on the data-driven estimates of NBP performed with the inversion of a global atmospheric transport model constrained by observations of atmospheric $CO_2$ concentrations. For this reason, the strengths and weaknesses of the study are related to those of the underlying NBP data product.

On the one hand, the atmospheric inversion technique offers the advantage for the specific goals of this assessment that the fluxes at any location are detected by the observational network, and can be spatially attributed on the large scales. That is, the results are not limited by an incomplete representation of ecosystems that may be inherent in estimates based on point level NBP observations. In addition, the inversion estimates cover more than 3 decades, representing the longest time series of spatially-explicit, observation-driven estimates of the terrestrial carbon fluxes. A third specific advantage of the JENA CARBOSCOPE inversion framework is that both the observation network (i.e. the number and location of stations of atmospheric stations) and the prior fluxes are constant during the simulation period. Consequently, temporal changes in the estimated NBP are most directly driven by the atmospheric concentration field.

On the other hand, the inversion estimates of ecosystem $CO_2$ fluxes are affected by uncertainties. Probably the largest source of uncertainty is transport model errors, in particular vertical mixing. Transport model errors are expected to affect mean fluxes and the amplitude of flux variations, but are likely also time-dependent themselves. Further, errors in the estimates of anthropogenic fluxes directly affect NBP estimates as the atmospheric signals reflect the total surface flux including anthropogenic emissions of $CO_2$. Additional limitation of the inversion estimates is that prior fluxes are generated with a land surface model (Sitch et al., 2003) which embeds a priori knowledge of the relationship between climate drivers and terrestrial $CO_2$ fluxes. As such, prior estimates may affect the mean sensitivities shown in Figures 1 to 3, while they don't affect the trends shown in the other figures given that the priors are mean annual climatology of modelled land fluxes and therefore do not show a temporal trend. Finally, inversion estimates cannot distinguish between the counteracting $CO_2$ fluxes originated from photosynthesis and respiration and can therefore provide only limited insights into the factors controlling the individual ecosystem processes. As a proxy, we therefore analysed NBP during the CUP and CRP that are dominated by photosynthesis and respiration, respectively. However, the signal in these two sub-periods is actually affected by both GPP and TER and therefore the results cannot be interpreted as they were originated by single processes. For instance the observed dominant role of light during CRP is suggesting that it is actually the light limitation of GPP that is controlling the rapid fluctuation of NBP also off-season.

The overall structural uncertainty of the JENA CARBOSCOPE was evaluated by comparing runs of the same inversion system performed with different number of atmospheric stations (and therefore temporal coverage). Further, uncertainties due to statistical assumptions in the a-priori error covariance structure were evaluated by varying the assumed de-correlation lengths or other covariance parameters.

## 4.2 Spatial patterns of climatic controls on NBP

The global distribution of the limiting factors of the net biome productivity shows a high level of spatial coherence, so that large regions are controlled by a specific environmental factor, varying with the climate background. The most common driver of the short-term fluctuations in NBP is radiation, with positive correlation in most of the Northern Hemisphere. This pattern is likely due to the favourable growing conditions in the temperate zone, where weekly to seasonal variations in the ecosystem $CO_2$ flux are controlled by light-limited GPP. On the contrary, negative correlations dominate in the Southern Hemisphere,

likely due to unfavourable growing condition during the sunny and dry season. Surprisingly also the northernmost latitudes show a negative correlation to radiation, suggesting a negative impact of sunny weather on the carbon budged, in line with recent findings about the reduction of NBP in the boreal zone, due to the anticipated phenology that reduces the uptake in summer (Buermann et al., 2018; Lian et al., 2020). This finding is of particular relevance since those regions are exposed to accelerated warming (IPCC, 2014) and store large quantity of carbon in terrestrial ecosystem (Carvalhais et al., 2014).

The second most important driver of short-term NBP fluctuations is temperature, with a positive correlation in most regions of the Southern Hemisphere at all the investigated temporal scale. This suggests that tropical ecosystems are still operating below their optimal temperature, as suggested by Huang et al., (2019). The sensitivity to soil water content show the expected strong positive control on NBP in warm and arid regions. A similar reduction in NBP due to soil moisture limitation and the non-linear response of carbon uptake to water stress was reported by Seddon et al. (2016) and agrees with what was observed

by Green et al., (2019) analysing outputs from four Earth system models. According to this study the most affected regions are those characterized by seasonally dry climate, like tropical savannahs and semi-arid monsoonal regions. A faster atmospheric $CO_2$ growth rate in drier periods is also reported in Humphrey et al. (2018), who conclude that drier years are associated with a weakening of the land carbon sink.

Our results differ substantially from the outcome of a previous study on the potential climatic constraint on NPP (Nemani et

al., 2003) based on monthly climate statistics and remote sensing observation of vegetation over two decades (1982-1999). In addition to the different methodology used in the two studies, it is important to stress that our assessment addresses NBP and therefore includes also $CO_2$ fluxes from heterotrophic respiration and disturbances, while the analysis by Nemani et al. was limited to primary productivity.

Additional insights on the environmental controls of NBP can be gained by assessing the fluxes for the periods when land is a

net sink or a net source of $CO_2$ (CUP and CRP, respectively). During CUP the strong control of temperature in the boreal zone is in accordance with a study performed on 23 FLUXNET sites that shows how variations in GPP at northern sites can be explained to a large extent by mean annual temperature (Reichstein et al., 2007). Rödenbeck et al. (2018), working at inter-annual time scale, found similar positive relationships of NBP and temperature during spring and autumn in all northern extra-tropical land areas, a signal which is consistent with photosynthesis being temperature limited in this time of the year. In

temperate regions the control of NBP during CUP is on the contrary led by radiation, whereas in the tropical zone by soil moisture.

The relative importance of radiation and temperature is reversed during the carbon release period, when the fluctuations in NBP are mostly controlled by the incoming radiation across most of the Planet. However, during CRP radiation limits NBP in opposite directions in the two hemispheres: positive dependence in the light-limited boreal CRP and negative dependence in the water-limited austral CRP. An important variation in the sign of temperature control occur between the CUP (positive relationship) and the CRP (negative control) (Figure 2&3). This pattern is likely due to the positive response of both photosynthesis and ecosystem respiration to increasing temperatures (Barr et al., 2007; Krishnan et al., 2008; Reichstein et al., 2002; Ueyama et al., 2014). This asymmetry in the thermal response of the $CO_2$ fluxes originated from photosynthesis and respiration is at the base of the large uncertainty of the terrestrial C budget under climate change (Friedlingstein et al., 2014).

## 4.3 Temporal variability of the key drivers

Robust and independent estimates of temporal changes in the limiting factors of NBP are particularly relevant, given the relevant changes in the climate drivers that occurred in the last three decades (IPCC, 2014) and the uncertainties on the ecosystem responses to varying climate drivers. The strongest signal emerging from the analysis is the broad increase in the positive sensitivity to radiation during both CRP and CUP in the Northern Hemisphere, while it is decreasing in most of the Southern Hemisphere where the average signal is negative. This positive trend observed in the last three decades is likely due to the increasing leaf area index (LAI) and primary productivity of the northern regions (Zhu et al., 2016), leading to increased light use efficiency and therefore to a stronger control of radiation on NBP. For the interpretation of these results it is important to consider that the ecosystem carbon exchange is controlled by light only in ideal growing condition, when neither temperatures nor water are limiting photosynthesis. The positive trend in sensitivity to solar radiation during CPU in the boreal zone can be therefore interpreted as a tendency toward improved growing conditions due to a reduction of low temperature limitations. From the positive trends in light sensitivity observed here one could infer that the recent changes in climate, $CO_2$ concentration and nutrient availability have eased the growing conditions of plants (Nemani et al., 2003).

Interestingly, the trend in sensitivity to radiation generally follows the sign of the mean sensitivity, with positive trends in climate regions characterized by positive sensitivity and vice-versa. This coherence between sensitivity and trend can be likely explained with the acceleration of the terrestrial carbon cycle that is inherently leading to an increased sensitivity of $CO_2$ fluxes to drivers. Ultimately this phenomenon is leading to an increased spatial variance in the response of ecosystem to radiation.

Concerning temperature, in Eurasia the sensitivity of NBP is decreasing with time, in agreement with Piao et al., (2017), who report a declining temperature response of spring NPP ascribed to reduced chilling during dormancy and emerging light limitation. The sensitivity to soil water content is mostly increasing, in particular during CRP in most regions, except Western Europe, in line with the recent findings by Buermann et al., (2018) about the increasing role of water limitation in the boreal zone. Soil water content shows an increasing control on the seasonality of NBP also in the US, South America and South Africa, confirming the increasing relevance of water stress on primary productivity (Humphrey et al., 2018; Jung et al., 2010) and control of arid zones on variability of the terrestrial carbon budget (Ahlstrom et al., 2015).

Finally, the analysis of the sources of variability of NBP revealed that the largest fraction of the signal is coming from the temporal variation in the ecosystem response to the environmental drivers and not from the variation of the drivers. Temporal variations in ecosystem responses may originate from structural and physiological changes in vegetation characteristics, eventually occurring in response to changing environmental conditions (Marcolla et al., 2011; Richardson et al., 2007). For instance, the large increase in the sensitivity to radiation could be due to the increase in LAI and subsequent increase in the fraction of absorbed radiation occurring in most of the Northern Hemisphere. Ultimately, indirect effects of climate on the ecosystem response to environmental drivers may amplify the overall impact of climate variability and trends on the future dynamic of the terrestrial carbon budget, posing further uncertainty on the efficacy and vulnerability of land-based mitigation strategies.

## 4 Conclusions

- We focused this analysis on the climate drivers of the sub-annual variability of land $CO_2$ fluxes, as derived from an atmospheric inversion system, in order to characterize the key driver in the different World regions and climates. The short-term drivers of NBP can be interpreted as the limiting factors of the ecosystem carbon budget at weekly to seasonal scale. The assessment of the dominant drivers and their temporal trends is essential to understand the potential impact of the changing climate on the terrestrial carbon budget, with the ultimate goal of reducing the large uncertainty about the role of land on the future climate trajectories (Friedlingstein et al., 2014).

- Given that the atmospheric inversion does not allow a direct separation of NBP in gross primary productivity and ecosystem respiration, we analysed two contrasting periods: the carbon uptake periods (CUP) when NBP is dominated by photosynthesis and the carbon release period (CRP) when NBP is dominated by respiration. Results show drastic differences in the response of the terrestrial carbon budget to environmental drivers in the two periods. More specifically, during the CUP we detected three clear driving factors, temperature in the northernmost regions, radiation in the temperate regions and soil water content in the tropical region, with temperature being the most common driver. During the CRP a large fraction of the planet is radiation-controlled, with positive correlation in the Northern Hemisphere and negative in the Southern. This contrasting pattern is likely due to the off-season light limited photosynthesis in the boreal hemisphere (triggering the positive correlation), and by the indirect negative effect of high radiation loads on photosynthesis in warm and arid regions of the southern hemisphere.

- The rapid changes in the climate drivers and in ecosystem properties observed in the last decades (e.g. greening) have driven important changes in the climatic control of the net biome productivity. In particular, air temperature shows a positive correlation with NBP in Eurasia, but with a decline in sensitivity over time; on the contrary, sensitivity to radiation is increasing almost in the entire Boreal Hemisphere both during CUP and CRP, suggesting that NBP is becoming increasingly light-limited at short-time scales.

-    Factoring out the sources of temporal variability of NBP we showed that ecosystem $CO_2$ fluxes are controlled more by the temporal variation in the ecosystem sensitivities to climate drivers than by the temporal changes of the drivers. This finding suggests that the indirect impacts of climate change on the ecosystem sensitivity may actually be more relevant than the direct impact of the climate variability on the terrestrial $CO_2$ fluxes. Ultimately, indirect climate effects may trigger an important amplification of direct climate impact on NBP, leading to unexpected and non-linear

responses.

      -    Overall this analysis shows the spatial complexity and the clear dependencies on the climate background of the environmental controls on the terrestrial carbon budget. The significant changes in the climate sensitivities occurred in the last three decades demonstrate the rapid, ongoing evolution of the relationships between climate and the terrestrial carbon budget. Advancing the knowledge on the limiting factors and their variation is an important step in

the understanding and predicting the impacts of climate change on the terrestrial carbon budget.

**Author contribution.** AC and BM conceived the study and designed the methods. CR provided the Jena Carboscope data. BM performed the data analysis. AC and BM interpreted the results. MM and CR contributed to the improvement of the methods and to the interpretation of results. AC and BM wrote the manuscript with contributions from the other co-authors.

**Competing interests.** The authors declare that they have no conflict of interest

**Data availability.**

The Jena Carboscope database can be found at http://www.bgc-jena.mpg.de/CarboScope/

WFDEI database at http://www.eu-watch.org/data_availability

ERA-Interim database at https://www.ecmwf.int/en/forecasts/datasets/reanalysis-datasets/era-interim

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

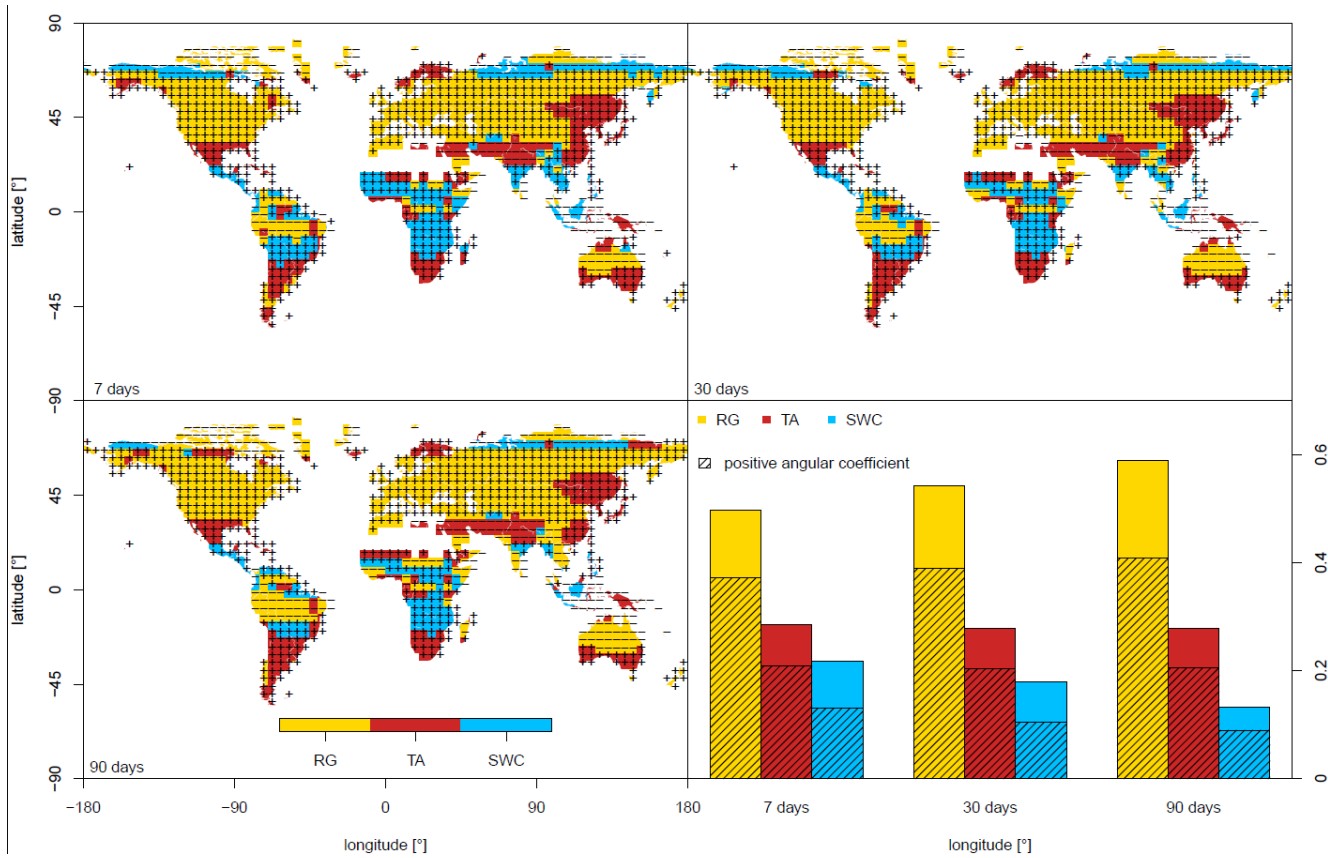

**Figure 1 - Maps of the dominant drivers calculated over the entire time series and sign of their angular coefficients in a multi-linear regression. Results are shown for three temporal resolutions, namely 7, 30 and 90 days. Bottom right panel: frequency of each dominant variable for the three analysed temporal resolutions, dashed areas represent the frequency of positive angular coefficients.**

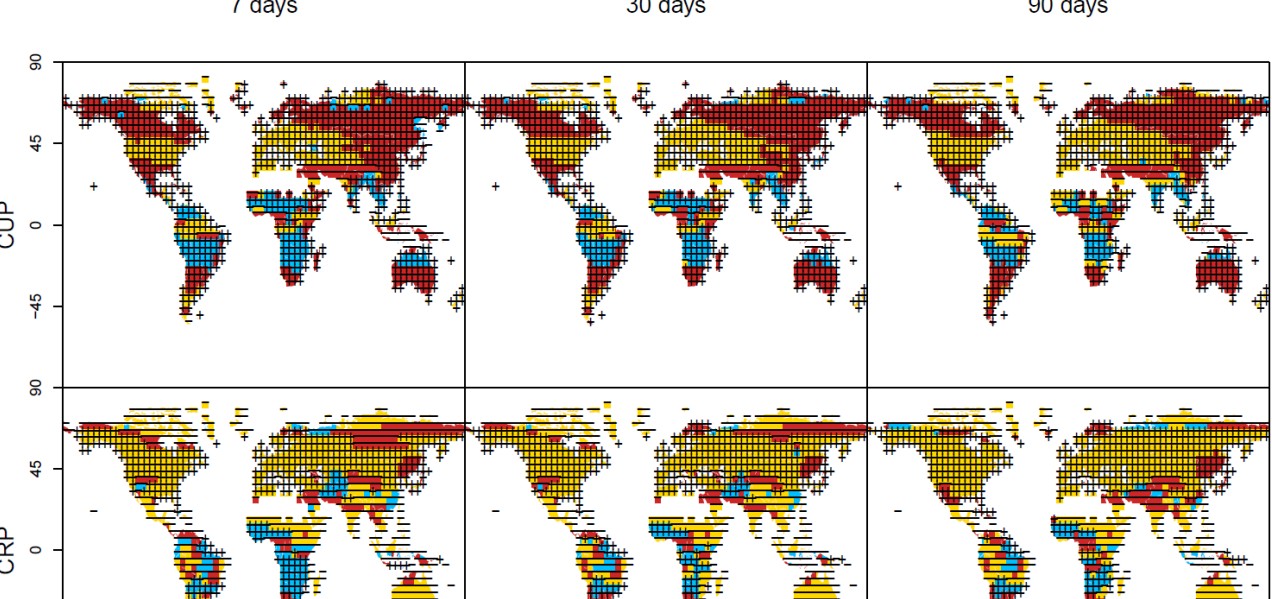

**Figure 2 - Maps of the dominant drivers and sign of their angular coefficients in a multi-linear regression calculated separately for Carbon Uptake Period (CUP, top row) and Carbon Release Period (CRP, bottom row). Results are shown for three temporal resolutions, namely 7, 30 and 90 days.**

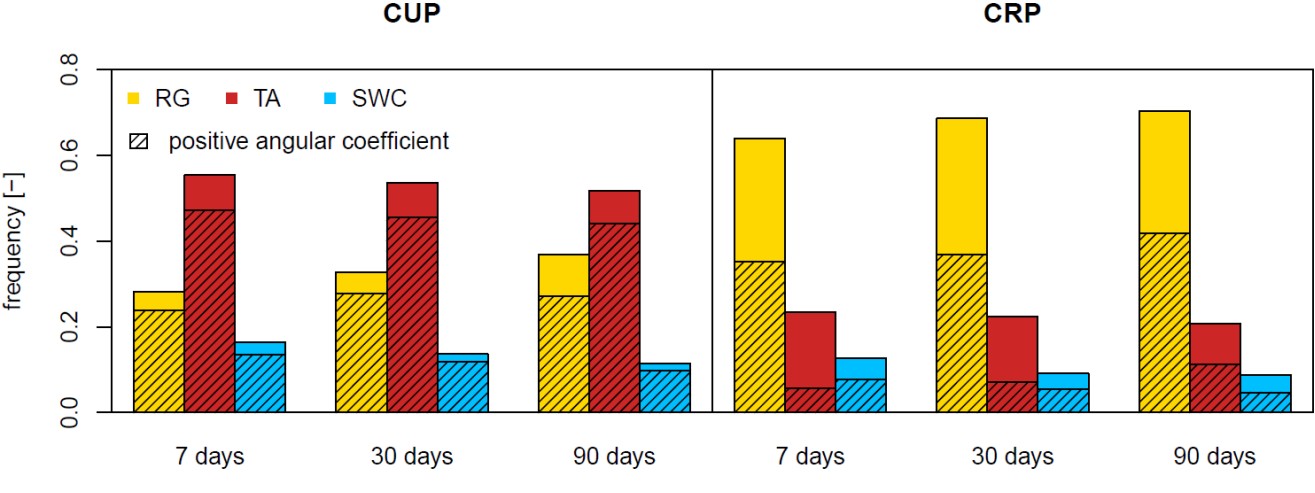


**Figure 3 - Frequency of the dominant variables plotted for Carbon Uptake Period (CUP) and Carbon Release Period (CRP) at different temporal resolutions (7, 30 and 90 days), and frequency of dominant variables with positive coefficients (dashed bars).**

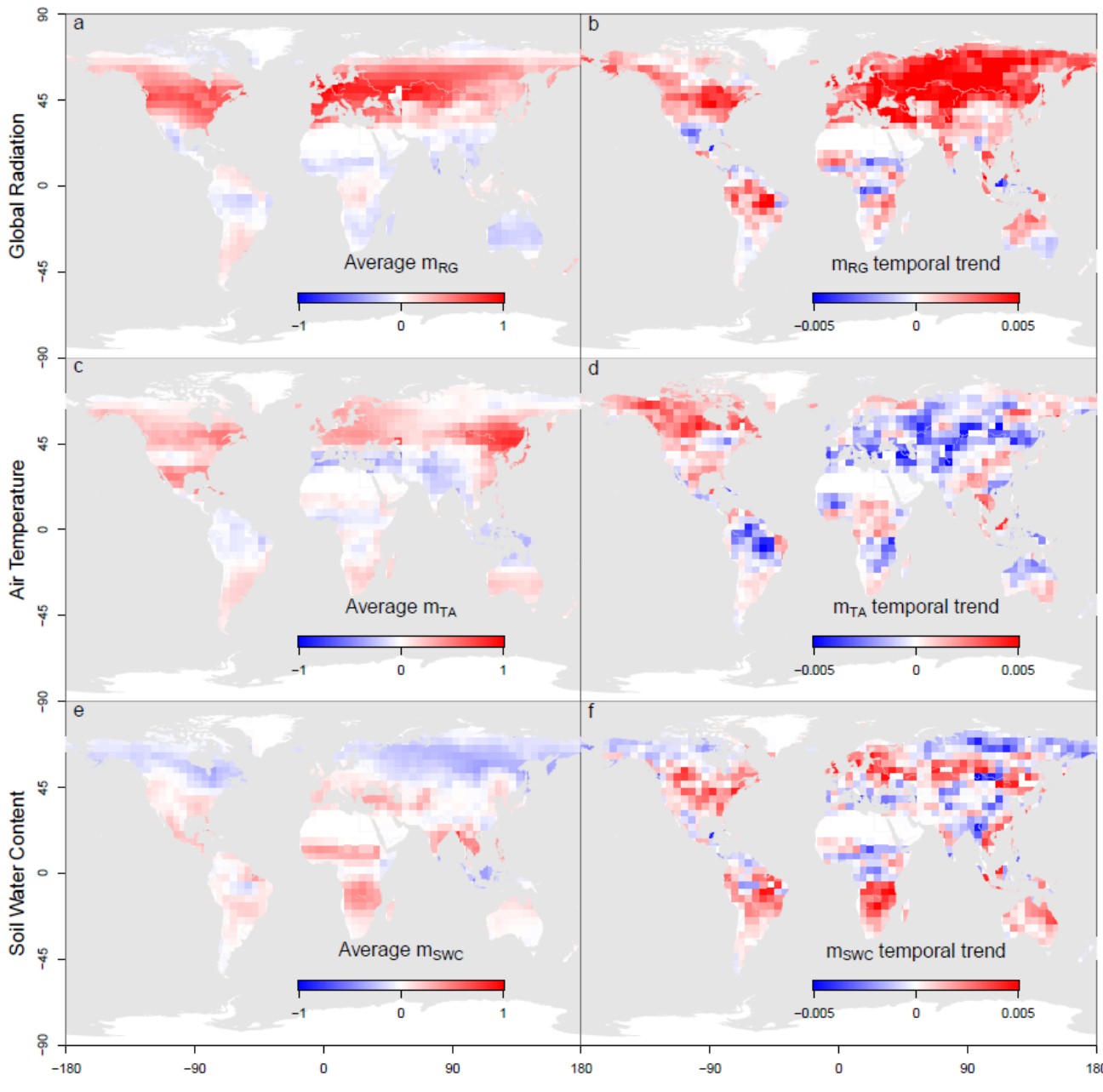


**Figure 4 – Maps of magnitude (left column) and trends (right column) of the sensitivity of Net Biome Productivity (NBP) to global radiation (first row), air temperature (second row) and soil water content (third row) at weekly time scale.**


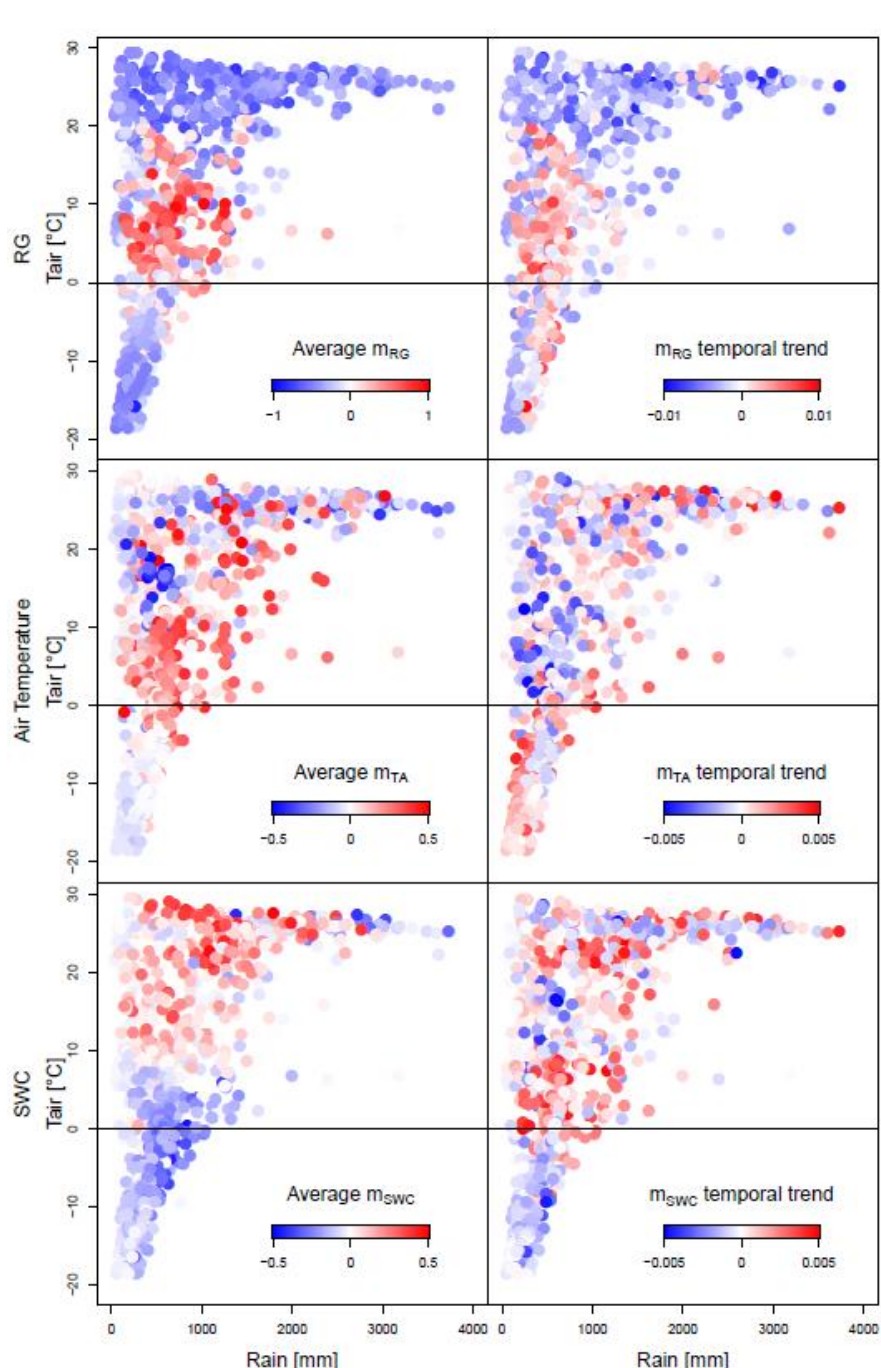

**Figure 5 – Scatter plots of sensitivity to climate drivers (left column) and of trends of the sensitivities (right column) plotted in a precipitation- temperature space at weekly time scale.**

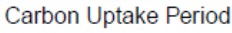

**Figure 6 – Maps of sensitivity temporal trends separately shown for Carbon Uptake Period (CUP, left column) and Carbon Release Period (CRP, right column) at weekly time scale. Maps are plotted for global radiation (first row), air temperature (second row) and soil water content (third row).**


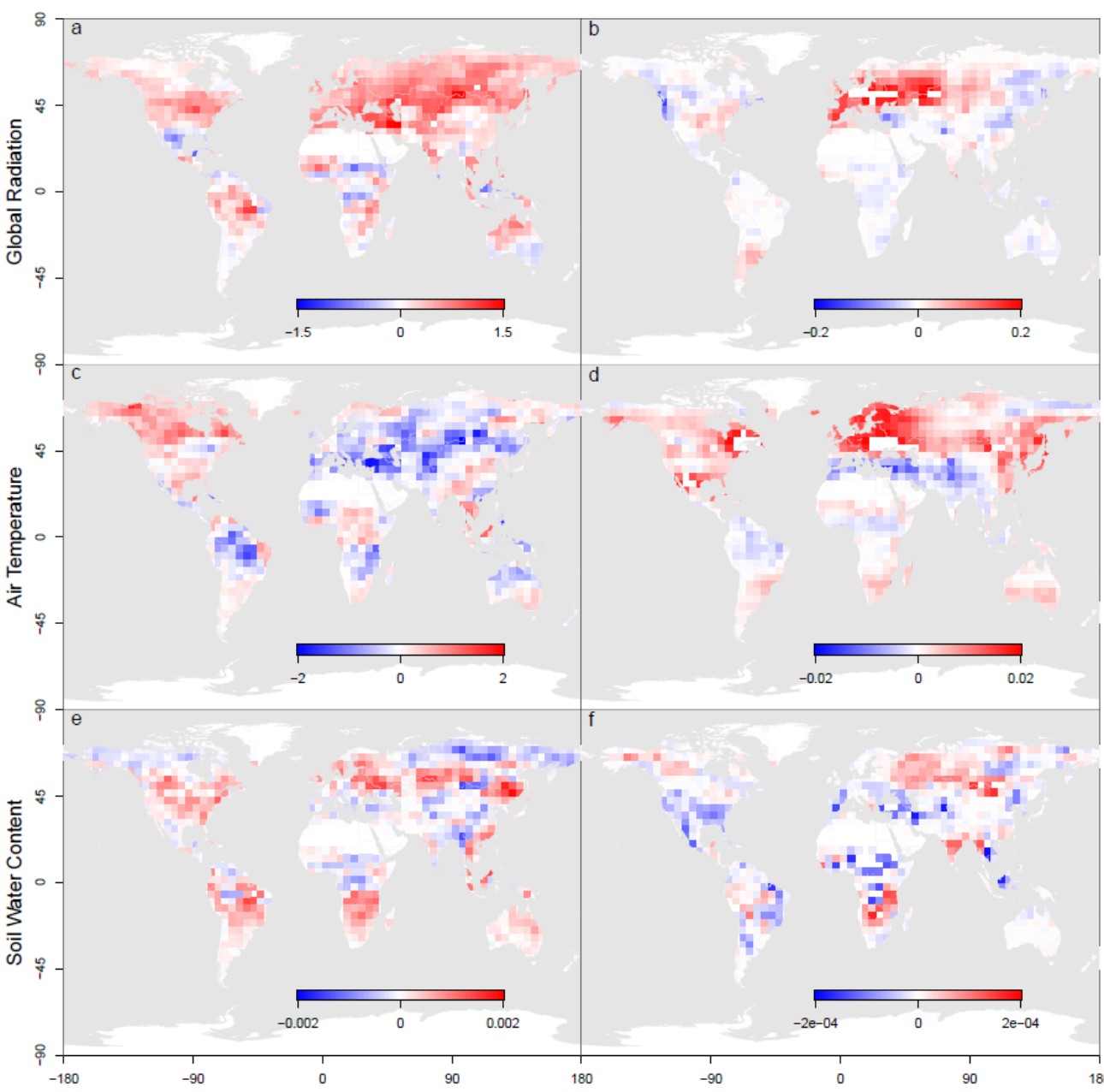


**Figure 7 – Maps of the average contribution of sensitivity temporal change (left column) and of the temporal change of the driver (right column) to the total temporal variability of Net Biome Productivity (NBP) in the investigated period. Maps are plotted for global radiation (first row), air temperature (second row) and soil water content (third row).**

