# Peer review of "Patterns and trends of the dominant environmental controls of net biome productivity"

_Biogeosciences, 2019_

## Referee Comment (RC1) · Anonymous Referee #1 · 24 Nov 2019

This paper investigated the global sensitivity of NBP to global radiation, temperature and soil water content from weekly to seasonal temporal scales(most at weekly scale) using one version of inversion NBP. What I concerned is the uncertainty of results because only one version of global NBP was used and the data had uncertainty at annual scale, particularly in weekly and monthly scale and the paper lacks the validation analysis, which make it unconvincing.

Generally, the abstract and introduction look good but the results and discussions are not good. The authors missed a lot of discussions and just simply describe the results. Throughout the manuscript, it should be more quantitative in nature.

Line 16-17, what datasets were used in this study?

[Figure]

line 20, how many are the relative contributions of radiation, temperature and soil water content?

Line 21 are you mean that soil water content plays a key role in arid regions of the southern hemisphere both in carbon uptake and release periods?

Line 23, the importance of radiation as a driver is increasing at global scale? Line 23, over what time period?

Line 23-24, So how many are the contribution of the temporal changes in ecosystem sensitivity and the temporal variability of the driver itself, respectively?

The title focused on NBP, but the it looks you are working on net ecosystem CO2 exchange (line 16) throughout the abstract. It should be specified rather than say carbon fluxes vaguely. Same problems in Introduction, you mentioned NBP in your questions but talked about NEE in the whole introduction.

Line 53-59, But it is at the hourly and daily scales where climate variability is directly acting on ecosystems too.

Line 53, The sensitivity of what?

Line 60-70, this paragraph is abrupt. It should be in Method. However, the specific climate factors also should be introduced in introduction before describing your aims.

Line 80, how many observed sites were used in this products? Different versions of Jena CarboScope CO2 Inversion have different numbers of observations and it is important to the uncertainty of NBP. Why do you chose the version s85_v4.1 rather than others? You only used the one product and version. This is a bit dangerous, how much can we trust your results?

Line 92, it should be better to include level 3 and 4, especially in forests and savannas. Line 102 what is the threshold of VIF used? Line 115, This is very dangerous because the inversion NBP may have large uncertainty at weekly and monthly scale for each

pix. So it is hard to convincing to define CUP and CRP.

Line 129, it is NBP, rather than net ecosystem CO2 exchange. Line 129, your abstract said the CO2 exchange over most of the land surface is controlled by temperature, but here you said it is radiation.

Figure 1, can you show the value for each drivers in the map rather than the dominant drivers simply? How can we know the positive or negative effect from this figure?

Line 136, summer drought decreases GPP but not increases TER. But radiation does not decrease GPP in the northernmost latitudes

Line 140, the reader don't know this number from this figure 1. I strongly recommend the author sperate the results and discussions because it is very unclear now. There are only two sentences in the some paragraphs of results. Line 142, As for radiation? Line 144, drier periods show higher uptake. Why? Line 153, so what? Line 158, the temperate zone is mostly radiation-driven. No, the temperate zone is mostly temperature-driven.

Line 161, but your results showed NBP is related to radiation and GPP is related to temperature. Line 164, are you taking about GPP, rather than NBP here? Line 165, The carbon release period of the Northern hemisphere is mostly driven by global radiation, which positively impacts on the NBP fluxes. So you mean carbon release period positively impacts on the NBP?

Line 170-172, how much is the positive or negative effect? Please add more quantitative descrbition. Figure 3, please show the frequency distribution curve. Line 173-176, are these differences between different drivers significant? Line 194, why? Line 199, why does an opposite positive trend of temperature sensitivity occur in North America? Line 200, which regions Line 206-208, these sentences should move to methods. Line 208, What clear pattern for radiation? Line 211-214, need to ref Figure 5 and 6, how about monthly and seasonal scale Line 238, you are not working on the weekly variation, rather than the inter-annual variability. Line 247, how bigger? I don't think you can compare them because you didn't normalize them together. Line 249 per se? Line 250, you need to compare this figure with greening map and see if it is true.

---

## Referee Comment (RC2) · Anonymous Referee #2 · 25 Nov 2019

This manuscript by Marcolla et al investigates global CO2 fluxes during the carbon uptake and carbon release period and at different time-scales. Overall, the paper is very interesting, the method sound and the manuscript well written. However, I did find that the discussion/broader impact was essentially missing, making it difficult to see what the consequences of this work are for the community.

Here are some comments: 1) The title " Recent changes in the dominant environmental controls of net biome productivity" is misleading. This paper does not look at "recent changes" or what the history of environmental controls was, so I would choose a title that reflects the actual paper better.

2) Section 2.2 is a little laborious, even though the actual analysis method is obvious once the reader gets to the figures. I would suggest illustrating the described analysis

with the evolution of a single pixel, it would help clarify the section.

3) Section 3 is a monstrous lock of text describing the figures one by one. The "Discussion" part of this section consists of a few sentences here and there. The paper would greatly improve if 1) The Section was split between "Results" and "Discussion" and 2) the "Results" section was split further into subsection for each type of analysis, just to help guide the reader through the overall progression of the analysis. I think that splitting the "Results" and "Discussion" would force the authors to put this work into perspective and draw conclusions about why this work matters for the different communities that might be interested in these results (flux tower, land surface modelers, global models, etc. . .).

4) In the Discussion section, it would also be helpful to include some limitations: how is the way vegetation is modeled influencing the results in one direction? Is the modeled know for modeling some aspects better than others? This would be a very valuable addition.

5) I would move Figures 3 to the Supporting Information since it doesn't actually show new data, just the same data from Figure 2 plotted differently. It is still nice to see though, so the SI would be a good place for it. Similarly, Figures 4 and 5 show essentially the same data. I found Figure 5 more interesting though, so I would again move Figure 4 into the SI.

Edits: overall, the text was very well written. My only minor comment on the text is that at line 142, I would replace "As for radiation" with "Similarly to radiation". The sentence is technically correct, but I found the use of "as" in this specific context to be confusing.

---

## Author Comment (AC1) · 11 Feb 2020

This paper investigated the global sensitivity of NBP to global radiation, temperature and soil water content from weekly to seasonal temporal scales(most at weekly scale) using one version of inversion NBP. What I concerned is the uncertainty of results because only one version of global NBP was used and the data had uncertainty at annual scale, particularly in weekly and monthly scale and the paper lacks the validation analysis, which makes it unconvincing.

Following the suggestion of the reviewer we repeated the analysis with other versions of the Jena Carboscope product that were specifically produced to explore the uncertainty of the inversion driven by the priors and the spatio-temporal correlation of the error (s81oc_v4.3, s85oc_tight_v4.3 with halved prior uncertainty, s85oc_loose_v4.3 with double prior uncertainty, s85oc_short_v4.3 with shorter spatial correlation and s85oc_fast_v4.3 with shorter temporal correlation). In addition, we explored also a previous version of the inversion based on a different number of atmospheric stations (s81oc_v4.3). We limited the analysis of the uncertainty to different versions of the Jena CarboScope $CO_2$ Inversion since, to our knowledge, all others long-term inversions are produced with a varying observation network (the number of atmospheric stations used in the inversion is changing during the time series) and are therefore not adequate for the scope of our study. In fact, temporal variations in the observation network generates spurious temporal trends in the signal driven by the additional observational constrains in the assimilation process.

The new analyses are fully consistent with the results reported in the manuscript for the large majority of the land surface. In the revised version of the manuscript we included new figures (which are attached below) to show the results of the uncertainty analysis and highlighting the consistency between products. In particular, Figure S1 shows the maps of the dominant drivers at three different time scales, black pixels are those for which less than 5 out of 6 inversion products agreed on the dominant driver selection. The bar-plot in Figure S1 shows that the results of 5 out of 6 products are consistent over about 90% of the land surface in terms of dominant driver. In addition, we repeated also the analysis on the temporal trend of the sensitivity with the other 5 inversion products. The first column of Figure S2 shows the average regression coefficients, the second column the standard deviation among the 6 products, while in the third column we plotted the sign of the trend of the regression coefficients; only pixels which showed an agreement in 5 out of 6 products in terms of sign were plotted in color, while black pixels are those for which less than 5 products agreed. These results were discussed in the manuscript and the figures were put in the supplementary material.

Generally, the abstract and introduction look good but the results and discussions are not good. The authors missed a lot of discussions and just simply describe the results. Throughout the manuscript, it should be more quantitative in nature.

Following the suggestion of the reviewer, we improved and deepened the discussion of the results in the updated version of the manuscript.

Line 16-17, what datasets were used in this study?

The acronym of the datasets has been added to the abstract. A full list and description of the datasets used in the study are reported in the Materials and Methods section.

line 20, how many are the relative contributions of radiation, temperature and soil water content?

Following the reviewer comment, we added the % of total pixels driven by temperature and radiation during CUP and CRP.

Line 21 are you mean that soil water content plays a key role in arid regions of the southern hemisphere both in carbon uptake and release periods?

Yes, according to results shown in Figure 2, SWC is the dominant driver of the arid regions of the Southern Hemisphere both during CUP and CRP. We made it clear in the new version of the manuscript

Line 23, the importance of radiation as a driver is increasing at global scale? Line 23, over what time period?

Figure 4 shows that the temporal sensitivity of radiation regression coefficient is positive in most of the Northern Hemisphere and in a large part of the Southern Hemisphere, which means that the increase of NPB at increasing radiation is increasing with time.
The time frame we are looking at is ~30 years. We clarified this point in the updated version of the manuscript

Line 23-24, So how many are the contribution of the temporal changes in ecosystem sensitivity and the temporal variability of the driver itself, respectively? The title focused on NBP, but the it looks you are working on net ecosystem CO2 exchange (line 16) throughout the abstract. It should be specified rather than say carbon fluxes vaguely. Same problems in Introduction, you mentioned NBP in your questions but talked about NEE in the whole introduction.

The contribution of temporal change in ecosystem sensitivity is about two orders of magnitude larger than the contribution of the temporal variability of the driver itself. The paper focuses on NBP since inversions cannot factor-out the $CO_2$ emissions driven by natural and anthropogenic disturbances. We changed the text where needed to be consistent.

Line 53-59, But it is at the hourly and daily scales where climate variability is directly acting on ecosystems too.

We agree with the remark of the reviewer, however working with the Inversion product at global scale it is not possible to go beyond the weekly time scale. We pointed this out in the manuscript in the Materials and Methods section.

Line 53, The sensitivity of what?

We specified "the sensitivity of ecosystem NBP to climate variability"

Line 60-70, this paragraph is abrupt. It should be in Method. However, the specific climate factors also should be introduced in introduction before describing your aims.

We agree with the reviewer and have updated the paragraph accordingly.

Line 80, how many observed sites were used in this products? Different versions of Jena CarboScope CO2 Inversion have different numbers of observations and it is important to the uncertainty of NBP. Why do you choose the version s85_v4.1 rather than others? You only used the one product and version. This is a bit dangerous, how much can we trust your results?

A network of 21 atmospheric sites were used for version s85. Following the reviewer remark we specified it in the text. Since the scope of our study was to investigate the temporal variation in the drivers we selected this version because it was a good compromise between time series length and robustness of the observation network. However, in order to test the dependency of the results on the product version used, we repeated part of the analysis with other versions of the Inversion product.

Line 92, it should be better to include level 3 and 4, especially in forests and savannas.

In ERA-Interim there is a high temporal correlation of the soil water content between levels so we don't expect relevant changes when using different levels. Given that we had to choose only one value for all vegetation types (at the spatial resolution of the inversion it is not possible to separate PFTs) we assumed that layer 2 (from 0.07 to 0.28 m depth) was the best representative of SWC for all ecosystem types (including woody and herbaceous PFTs).

Line 102 what is the threshold of VIF used?

According to the literature on the subject a VIF value of 5 was considered as a threshold for multi-collinearity. Maps of VIF at three temporal resolutions have been added in the supplementary information. As expected RG and TA show high collinearity, VIF values increase at decreasing time resolution, only SWC shows VIF<5 over most of the land surface.

Line 115, This is very dangerous because the inversion NBP may have large uncertainty at weekly and monthly scale for each pix. So it is hard to convincing to define CUP and CRP.

We agree with the reviewer on the issue of the uncertainty for inversion retrievals at the scale of the single grid-cell; however we believe that the uncertainty leads mostly to random errors that should not mine the validity of the results when derived from a large number of grid-cells at the global scale. In fact, despite these uncertainties, our analysis shows coherent patterns across geographic regions for the CUP/CRP analysis, therefore suggesting that our sensitivities metrics are robust.

Line 129, it is NBP, rather than net ecosystem CO2 exchange. Line 129, your abstract said the CO2 exchange over most of the land surface is controlled by temperature, but here you said it is radiation.

We agree with the reviewer and therefore throughout the text we checked for consistency and changed from NEE to NBP where needed. The sentence at line 129 refers to Figure 1 in which results are shown for the whole time series without a distinction into CUP and CRP. The sentence in the abstract is related to the results of Figure 2 in which CUP and CRP are separately analyzed. What we observe is that radiation controls the sub-annual fluctuations of NBP in most of the northern hemisphere, while when the NBP time series is separated into CUP and CRP, radiation is still the most frequent dominant driver during CRP and temperature is the most frequent dominant driver during CUP.

Figure 1, can you show the value for each drivers in the map rather than the dominant drivers simply? How can we know the positive or negative effect from this figure?

The values of regression coefficients for each driver are shown in Figure 4 (left panels) for the weekly time scale. The symbols plus and minus overlaying the color map refer to the sign of the regression coefficient of the dominant driver, hence when a plus is plotted over a pixel it means that in that pixel the dominant driver has a positive impact on NBP.

Line 136, summer drought decreases GPP but not increases TER. But radiation does not decrease GPP in the northernmost latitudes

We agree with the reviewer and changed the sentence as follows "Surprisingly also the northernmost latitudes show a negative correlation to radiation, suggesting a negative impact of sunny weather on the carbon budged, in line with recent findings about the reduction of NBP in the boreal zone, due to the anticipated phenology that reduces the uptake in summer".

Line 140, the reader don't know this number from this figure 1. I strongly recommend the author sperate the results and discussions because it is very unclear now. There are only two sentences in the some paragraphs of results.

The percentages reported at line 140 can be retrieved form the bar plot of Figure 1. Following the reviewer suggestion we separated the results and discussion sections and improved the discussion of the results.

Line 142, As for radiation?

We changed it into "Similarly to radiation…"

Line 144, drier periods show higher uptake. Why?

Our interpretation of this result is that humid/rainy periods at the northernmost latitudes are characterized by a combination of low radiation and low temperature, which may ultimately limit primary productivity. Soil water content controls the boreal latitudes and has a negative effect on the carbon fluxes; while in arid regions of the Southern Hemisphere it has a positive effect (humid periods show higher $CO_2$ flux).

Line 153, so what?

In the revised version of the manuscript we described in further details the differences between our study and the one by Nemani et al. (2003) which is based on remote sensing retrievals of vegetation indexes to estimate NPP and therefore not accounting for heterotrophic respiration.

 Line 158, the temperate zone is mostly radiation-driven. No, the temperate zone is mostly temperature-driven.

Our results show that the short term variations of NBP in the temperate zone is radiation driven during CRP and show a mixed pattern of temperature and radiation limitation during CUP (Figure 2). We changed the sentence at line 158 accordingly.

Line 161, but your results showed NBP is related to radiation and GPP is related to temperature.

Figure 2 shows that NBP during CUP (GPP proxy) in the boreal region is actually controlled by temperature in accordance to Reichstein et al (2007).

Line 164, are you taking about GPP, rather than NBP here?

We are talking about NBP during CUP, which is used as a proxy for GPP.

Line 165, The carbon release period of the Northern hemisphere is mostly driven by global radiation, which positively impacts on the NBP fluxes. So you mean carbon release period positively impacts on the NBP?

What we observe is that radiation is the dominant driver in most of the Northern Hemisphere and that sensitivity to radiation (its regression coefficient) is positive, which means that NBP increases at increasing radiation at the investigated temporal scales.

Line 170-172, how much is the positive or negative effect? Please add more quantitative descrbition.

The absolute magnitude of the sensitivity for the different drivers is reported in Figure 4. The image shows that on average the positive sensitivity to SWC is higher than the negative sensitivity to radiation.

Figure 3, please show the frequency distribution curve.

In Figure 3 we show the frequency distribution of the dominant drivers at the investigated time scales separately for CUP and CRP.

Line 173-176, are these differences between different drivers significant?

We performed a CHI-squared test which proved that the distributions are statistically different.

Line 194, why?

Negative correlations dominate in the Southern Hemisphere, likely due to unfavorable growing condition during the sunny and dry season (that explain the average negative sensitivity) and the large spatial variation in the terrestrial water budget (leading to the heterogeneity in the trends).

Line 199, why does an opposite positive trend of temperature sensitivity occur in North America?

We could not find a robust explanation for this pattern.

Line 200, which regions

Evaporation is supply limited from temperate to Mediterranean and tropical arid regions, while demand limited regions are located in boreal arctic and humid tropics.

Line 206-208, these sentences should move to methods.

The sentence is meant as a link between Figure 4 and Figure 5 description. We would therefore prefer to keep it at the current location.

Line 208, What clear pattern for radiation?

We observed negative sensitivities in regions with high and very low temperature independently from precipitation values, while at intermediate temperatures it has a positive effect on NBP; and this holds also for the temporal trend of the sensitivity. This pattern and its potential causes is resumed and discussed in the revised version of the manuscript.

Line 211-214, need to ref Figure 5 and 6, how about monthly and seasonal scale

Following the reviewer suggestion we performed the analysis also at monthly and seasonal scale; Figures are reported in the supplementary material.

Line 238, you are not working on the weekly variation, rather than the inter-annual variability.

The sentence was reworded as follows: "Soil water content shows an increasing control on the seasonality of NBP also in the US South America and South Africa, confirming the increasing relevance of water stress on primary productivity (Jung et al., 2010) and control of arid zones on variability of the terrestrial carbon budget (Ahlstrom et al., 2015)."

Line 247, how bigger? I don't think you can compare them because you didn't normalize them together.

The contribution of the change in sensitivity to driver is almost two orders of magnitude larger that the contribution due to the temporal change of the driver itself. These two contributions sum up to build the total NBP temporal change (see equation at line 134).
The two terms have the same units and are comparable. They contribute to build up the total temporal variation of NBP and their contribution was disentangled and reported in Figure 7 of the manuscript.

Line 249 per se?

The wording does not appear in the new version of the manuscript.

Line 250, you need to compare this figure with greening map and see if it is true.

Following the suggestion of the reviewer we added in the text a comment about the match of our maps with that of the greening with additional references.

[Figure]

**Figure 1s: maps of the Variance Inflation Factor (VIF)**

[Figure]

**Figure 2s: maps of the dominant drivers calculated over the entire time series. Results are shown for three temporal resolutions, namely 7, 30 and 90 days. Black pixels are those for which less than 5 out of 6 inversion products agreed on the dominant driver selection. The bar-plot in Figure 1s shows that the frequency of pixel for which a certain number of products agree on the dominant driver selection. Outcomes of 5 out of 6 products are consistent over about 90% of the land surface.**

[Figure]

**Figure 3s: Maps of magnitude (first column) of the sensitivity (m) of Net Biome Productivity (NBP) to global radiation (first row), air temperature (second row) and soil water content (third row), maps of the standard deviation (second column) of m between products, sign of the temporal trend of m (third column) at weekly time scale. In the third column only pixels which showed an agreement in 5 out of 6 products in terms of sign were plotted in color, while black pixels are those for which less than 5 products agreed.**

[Figure]

**Figure 4s: same as Figure 4 in the main text, but for the 30 day time scale**

[Figure]

**Figure 5s: same as Figure 4 in the main text, but for the 90 day time scale**

---

## Author Comment (AC2) · 11 Feb 2020

This manuscript by Marcolla et al investigates global CO2 fluxes during the carbon uptake and carbon release period and at different time-scales. Overall, the paper is very interesting, the method sound and the manuscript well written. However, I did find that the discussion/broader impact was essentially missing, making it difficult to see what the consequences of this work are for the community. Here are some comments:

1)The title " Recent changes in the dominant environmental controls of net biome productivity" is misleading. This paper does not look at "recent changes" or what the history of environmental controls was, so I would choose a title that reflects the actual paper better.

Following the reviewer suggestion we changed the title into:

"Patterns and trends of the dominant environmental controls of net biome productivity"

We would like to keep the focus also on the temporal dynamics of the controls since this is a relevant goal of the work (see Fig. 4, 5, 6).

2) Section 2.2 is a little laborious, even though the actual analysis method is obvious once the reader gets to the figures. I would suggest illustrating the described analysis with the evolution of a single pixel, it would help clarify the section.

We reworded Section 2.2 in order to better clarify the applied methodology.

3) Section 3 is a monstrous lock of text describing the figures one by one. The "Discussion" part of this section consists of a few sentences here and there. The paper would greatly improve if 1) The Section was split between "Results" and "Discussion" and 2) the "Results" section was split further into subsection for each type of analysis, just to help guide the reader through the overall progression of the analysis. I think that splitting the "Results" and "Discussion" would force the authors to put this work into perspective and draw conclusions about why this work matters for the different communities that might be interested in these results (flux tower, land surface modelers, global models, etc. . .).

Following the reviewer's suggestion we separated Results and Discussion into two separate sections. We focused the Results on the most relevant findings and and improved the Discussion section.

4) In the Discussion section, it would also be helpful to include some limitations: how is the way vegetation is modeled influencing the results in one direction? Is the modeled know for modeling some aspects better than others? This would be a very valuable addition.

We agree with the reviewer on this point and we have therefore added a first section in the discussion on the limitation of the method.

5) I would move Figures 3 to the Supporting Information since it doesn't actually show new data, just the same data from Figure 2 plotted differently. It is still nice to see though, so the SI would be a good place for it. Similarly, Figures 4 and 5 show essentially the same data. I found Figure 5 more interesting though, so I would again move Figure 4 into the SI.

We think that the bar plot of Figure 3 contains an additional information which is not evident from Figure 2, i.e. the frequency change across temporal scales and this is the reason why we would prefer to maintain the figure in the main text. We agree with the reviewer that Figure 4 and 5 show the same results but figure 4 gives the spatial information which is lost in Figure 5 where results are plotted in climate coordinates.

Edits: overall, the text was very well written. My only minor comment on the text is that at line 142, I would replace "As for radiation" with "Similarly to radiation". The sentence is technically correct, but I found the use of "as" in this specific context to be confusing.

The sentence was changed accordingly to the reviewer suggestion